# Relevance of reporting leprosy related disability at the completion of multi drug therapy: A 5-year retrospective analysis of disability in persons affected by leprosy at ALERT Hospital Ethiopia

**Bereket Abebayehu Tegene**[1]*, **Thomas Asfaw Atnafu**[2,3]

**1** Department of Clinical Research, London School of Hygiene and Tropical Medicine, London, United Kingdom, **2** Department of Internal Medicine, Addis Ababa University, Addis Ababa, Ethiopia, **3** All Africa Leprosy TB and Rehabilitation Training Centre (ALERT), Addis Ababa, Ethiopia

* abebayehu.bereket25@gmail.com

**Data Availability Statement:** All relevant data are in the manuscript and it's supporting information files.

## Abstract

### Background

Leprosy is one of the neglected tropical diseases associated with significant morbidity in endemic regions. It causes disability affecting the daily activities and social participation of affected individuals. Understanding the prevalence and trend of leprosy-related disability throughout the world and the accuracy of disability data counted by WHO is crucial in guiding efforts to be made towards the targets set by WHO to be achieved by 2030. This study aims to show the significance of reporting leprosy-related disability at the end of MDT and critique how disability is counted in the context of WHO data.

### Methods

This is a mixed method study with a 5-year retrospective analysis of outcomes of newly diagnosed leprosy patients at ALERT Hospital in Ethiopia from 2016 to 2020. A comparative review and analysis of leprosy related G2D (Grade 2 Disability), globally, regionally, and in Ethiopia using WHO data was also done. In addition, semi-structured interview of health workers (HCWs) and professionals working in the field of leprosy at various organizations was conducted.

### Results

The trend of G2D among newly diagnosed leprosy patients shows no decline globally for the past 20 years. It is increasing in Africa and stable in the Southeast Asian and American regions where majority of leprosy patients are found showing the gap in early case identification and prompt treatment of leprosy cases. The total number of newly diagnosed leprosy cases at ALERT hospital between January 2016 and December 2020 were 1032 and among those patients who had completed treatment the prevalence of G2D was 33% at

**Funding:** The author(s) received no specific funding for this work.

**Competing interests:** The authors have declared that no competing interests exist.

diagnosis and 23% at completion. The interview has also shown gaps in the completeness and quality of disability data reported to WHO and how disability is counted.

## Conclusion

Leprosy related G2D among newly diagnosed patient is not declining worldwide and even increasing in endemic regions like Ethiopia. More training should be given to health professionals in assessing disability. WHO should make some changes in the way it counts disability as the current definitions are prone to interpretation bias and lacks uniformity among various programmes and health workers. Prospective studies are needed in assessing disability progression post MDT so as design interventions and strategies in preventing worsening of disability after patients are discharged from treatment centre.

### Author summary

Leprosy is a bacterial infection caused by the Mycobacterium leprae. It mainly affects the skin, nerves and eyes leading to loss of sensation and dryness of the skin, blindness and deformity of fingers of the hands and feet. These visible physical health problems on affected individuals limits their daily activities and impedes their social interaction due to associated stigma and discrimination. The authors of this study examined a five-year data of newly diagnosed leprosy patients at ALERT Hospital in Ethiopia between 2016 and 2020.Disability caused by leprosy in Ethiopia, regionally and globally was assessed. In addition, interview was conducted with health professionals and leprosy experts. The number of patients with Grade 2 Disability, the highest level of leprosy disability, is not decreasing worldwide. At ALERT Hospital, 33% of diagnosed patients had Grade 2 Disability. The interviews also revealed gaps in disability counting and reporting. This study highlights that disability among newly diagnosed leprosy patients is not decreasing globally and even increasing in countries like Ethiopia where the disease is common. Therefore, more emphasis should be given in equipping health professionals in properly evaluating disability and standard definitions in assessing and counting disability should be used across leprosy programmes.

## Introduction

Leprosy is a chronic infectious neglected tropical disease caused by the acid-fast bacillus Mycobacterium leprae. Leprosy mainly affects the skin, nerves, and eyes commonly presenting with skin lesions and loss of sensation and motor impairment due to peripheral nerve damage [1]. Eye involvement occurs in almost 75% of cases and blindness occurs in 5% of cases [1]. It can also affect the autonomic nervous system leading to dryness of the skin. The nerve damage leading to sensory loss makes affected persons vulnerable to repeated injuries leading to deformity and loss of digit while involvement of motor nerves causes nerve trunk enlargement and motor weakness causing wrist and foot drop and claw hands and feet [1,2].The diagnosis of leprosy is mainly clinical with or without slit-skin smear examination. It can be diagnosed based on the presence of at least one of the cardinal signs: (i) definite loss of sensation in a pale (hypopigmented) or reddish skin patch; (ii) thickened or enlarged peripheral nerve with loss of sensation and/or weakness of the muscles supplied by that nerve; or (iii) presence of acid-

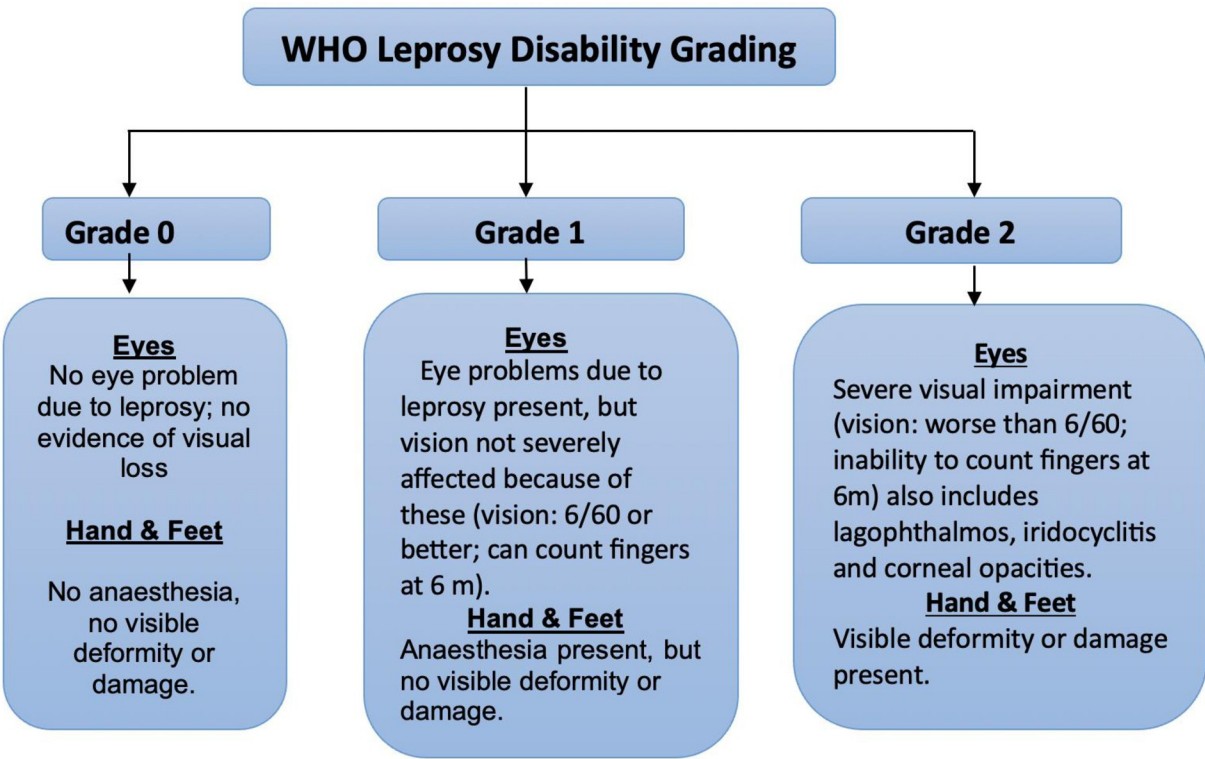

**Fig 1. Flow chart of WHO Leprosy Disability Grade [11].**

fast bacilli in a slit-skin smear [3]. The World Health Organization (WHO) classifies leprosy into two types based on the number of lesions. If there are five or fewer lesions, it is classified as paucibacillary (PB) case and Multibacillary (MB) if there are more than five lesions and this classification is used for treatment purposes [4].Currently WHO recommends a 3-drug regimen which consists of rifampicin, dapsone and clofazimine for all leprosy patients with 6-month of treatment for patients with PB leprosy and 12 month for patients with MB leprosy [3]. Leprosy reactions are immune mediated inflammatory responses to M. leprae antigen and occurs before, during treatment or after treatment of MDT with two distinct types known as Type 1 reactions (T1Rs) and Type 2 reactions (T2Rs).These reactions cause disability through peripheral nerve damage and patients present with swelling of the face, hand and feet, motor and neurologic impairment, inflamed skin lesions challenging patients' adherence to treatment especially if they are not well informed. It also requires long term immunosuppressive therapy [5–7].There is also increased risk of reactions during MDT treatment [8].

Globally the number of leprosy cases on treatment at the end of 2021 was 133,802, and the prevalence rate was 16.9 per million population. In the same year the number of new cases reported in the respective WHO region were as follows: African-21,201; Americas-19,826;Eastern Mediterranean-3,588;European-14;Southeast Asia-93,485 and Western pacific -2480 [9].

The physical impairments due to the disease lead to disabilities causing hinderance to activities involving the hands, feet, and eyes. WHO categorizes leprosy related impairments into three grades: Grade 0-no impairment (G0D), Grade 1-loss of sensation in the hand or foot (G1D) and Grade 2-visible impairment or deformity (G2D) [10].Eyes, hands and feet are graded separately and receive a score of 0,1 and 2(*Fig 1*).All six scores should be graded and the grade of the person as a whole is the highest score in any of the six places [11]. Sensation

testing for light touch can be done using either of ballpoint pen, a single monofilament or a set of graded monofilaments [7].

Physical disability can occur before leprosy is diagnosed, during treatment or after MDT is completed. Further worsening of disability following completion of MDT treatment and once patients are declared cured is an issue since patients are not routinely followed after they are released from treatment centre [12].The WHO estimated that currently 3–4 million people are living with disability due to leprosy [13].

The WHO leprosy disability grading system was initially developed to be used by WHO as an indicator for early case detection and estimation of the delay based on the number of new cases with G2D at diagnosis while HCWs have using it as a change indicator for assessing progression in physical impairment and disability while patients are on treatment [11].National leprosy services use this grading system as an epidemiological indicator for efficacy of public health programmes [14]. Furthermore, is has been used to assess the effectiveness of care on disability grade improvement or deterioration after diagnosis and initiation of MDT [11].

The aim of this study is to show the prevalence and trend of leprosy related disability and critique the reliability and usefulness of WHO Leprosy related disability data based on the data from ALERT hospital in Ethiopia.

## Methods

### Ethics statement

Ethical approval was obtained from the from the MSc research ethics committee of the London School of Hygiene and Tropical Medicine (LSHTM Ethical approval reference number-27493) and Local ethical approval was obtained from AAERC (ALERT/AHRI Ethical Review Committee-PO/24/22). Formal consent was obtained from all health professionals and leprosy experts who participated in the interview.

### Study design

This research used a mixed method study design. Initially analysis of published WHO leprosy related G2D data including in children globally, regionally and in Ethiopia using the Weekly Epidemiological Record (WER) was done.The quantitative part is a retrospective analysis of newly diagnosed leprosy patients at All Africa Leprosy, Tuberculosis and Rehabilitation Training Centre (ALERT) hospital in Ethiopia from January 1st, 2016, to December 31st, 2020. Data was extracted and analysed from patient registry book and prevalence of WHO disability grade was estimated and logistic regression was done to describe factors associated with disability presence at diagnosis of leprosy patients who completed MDT at ALERT hospital. The qualitative part is a semi structured interview of HCWs (health care workers) and professionals working in leprosy programmes. Interviews were conducted using questions developed based on the findings from ALERT hospital and the analysis of WER reports of WHO on leprosy to understand the gaps and utility of WHO disability data.

### Study settings

The patient data was collected from ALERT hospital in Addis Ababa, the capital city of Ethiopia. It has been serving as a referral and teaching hospital with more than 240 beds and consists of dermatology, ophthalmology, surgery, and orthopaedics departments. It is the largest referral hospital in the country known for providing diagnostic and treatment services for persons affected by leprosy. The leprosy treatment unit on average sees around 470 newly diagnosed patients annually and provides service for patients coming from all regions of the country by

trained health workers on leprosy and record their data using standard leprosy registry books which are used throughout the country which includes patient's name, age, sex, address, smear result, category, disability grade at diagnosis and completion of treatment, total number of house hold contacts and diagnosed with leprosy, date of start and end of treatment, progress at completion of MDT and corticosteroid use.

## Study population

All patients newly diagnosed with leprosy at ALERT hospital during the specific period were included and there were no exclusion criteria used. Health care workers (nurses and physicians) with more than 2 years of experience working in the leprosy clinic at ALERT hospital were interviewed. The remaining professionals were individuals purposively selected with extensive experience working on leprosy at local, government and international organizations.

## Data collection

Anonymised data was extracted from leprosy registry books at ALERT hospital and entered into data collection form using the ODK (Open Data Kit) application and encrypted and stored on the ODK central server using LSHTM Global Health Analytics ODK system. The data included in this study were sociodemographic (age, sex, region) and clinical features including smear result, category, WHO disability grade at diagnosis and at completion of MDT, disability grade condition (improved, same or deteriorated) at completion and steroid use with reasons for use.

Semi structured interview with HCWs and professionals working in the field of leprosy were conducted remotely in English via after informed consent was taken and was based on questions tailored according to the professional level of the interviewee. The HCWs were interviewed about issues related with leprosy associated disability data completeness in recording and reporting while the other professionals' interview was focusing on problems in disability data reporting to WHO and national leprosy programme, significance of Grade 1 disability recording and reporting, opinions about collecting disability data at completion of MDT treatment and post MDT and challenges with health care workers in leprosy clinics. The interview questions were initially pilot tested and standardized to reduce any form of bias.

## Data management and analysis

The prevalence trend of grade 2 disability at global, regional, and national level was analysed. The collected data from ALERT hospital was checked for completeness on a daily basis by going through the ODK portal and looking if it is appropriately filled. Data were stored on the central server of LSHTM and was imported and analysed using STATA 17.0 (Stata Corp LP). Descriptive statistics was done to calculate the percentage of disability data completeness in ALERT hospital and prevalence of disability grade. Bivariate and multivariate logistic regression was undertaken to calculate the crude and adjusted odds ratio with 95% confidence interval and significance level of 5% to identify factors associated with presence of disability at diagnosis. Gender, age, leprosy type(category), smear result and bacteriologic index were included in the multivariate analysis.

The audio files of the interview were anonymized and securely stored on the LSHTM server and in a personal computer in which it was deleted at the end of the study. The audio recording of the interview was transcribed and manually analysed through thematic analysis and themes were identified during the analysis.

## Result

### Analysis of the trend of leprosy related G2D

The percentage of G2D among newly diagnosed leprosy globally had been increasing from 15,074(3.77%) in 2004 until 2015 to be 14,059 (6.84%) in which it starts to show minimal decrement afterwards. The prevalence of G2D among new cases was highest in the Eastern Mediterranean region, which includes 22 countries in West Asia, North Central Asia, the Horn of Africa(Except Ethiopia and Eritrea),had been increasing since 2004 until 2011 in which it starts to decline afterwards until 2020 [15]. In the African region the trend of G2D has shown increase until 2019. In the western pacific region G2D ranging between 10 and 12 percent until 2012 and the starts to drop until 2017 where it starts to increase and level off until 2020. In the American region the prevalence of G2D ranges between 6 and 8 percent with minimal variation in between. Southeast Asian region has the lowest prevalence of G2D throughout those years. Globally the prevalence of G2D ranging between 4 and 6 percent with minimal variation (*Fig 2*).

The trend of G2D among newly diagnosed leprosy cases in Ethiopia was 776(14.5%) in 2004 and dropped to 692(6.8%) in 2005 and rises back to 589(10.7%) in 2006 and remains similar for the following two years and dropped to reach 260(6.9%) in 2012 and starts increasing for the next 5 consecutive years to peak in 2017 reaching 402(12.9%). In 2018 it gets lower to be 68(8.0%) and then increases until 2020. *(Fig 3)*

WHO has started recording the percentage of G2D among newly diagnosed children since 2016.The trend of G2D among newly diagnosed child cases globally was 281(1.52%) in 2016 where it had been increasing every year to reach 308(3.56%) in 2020 (*Fig 4*).

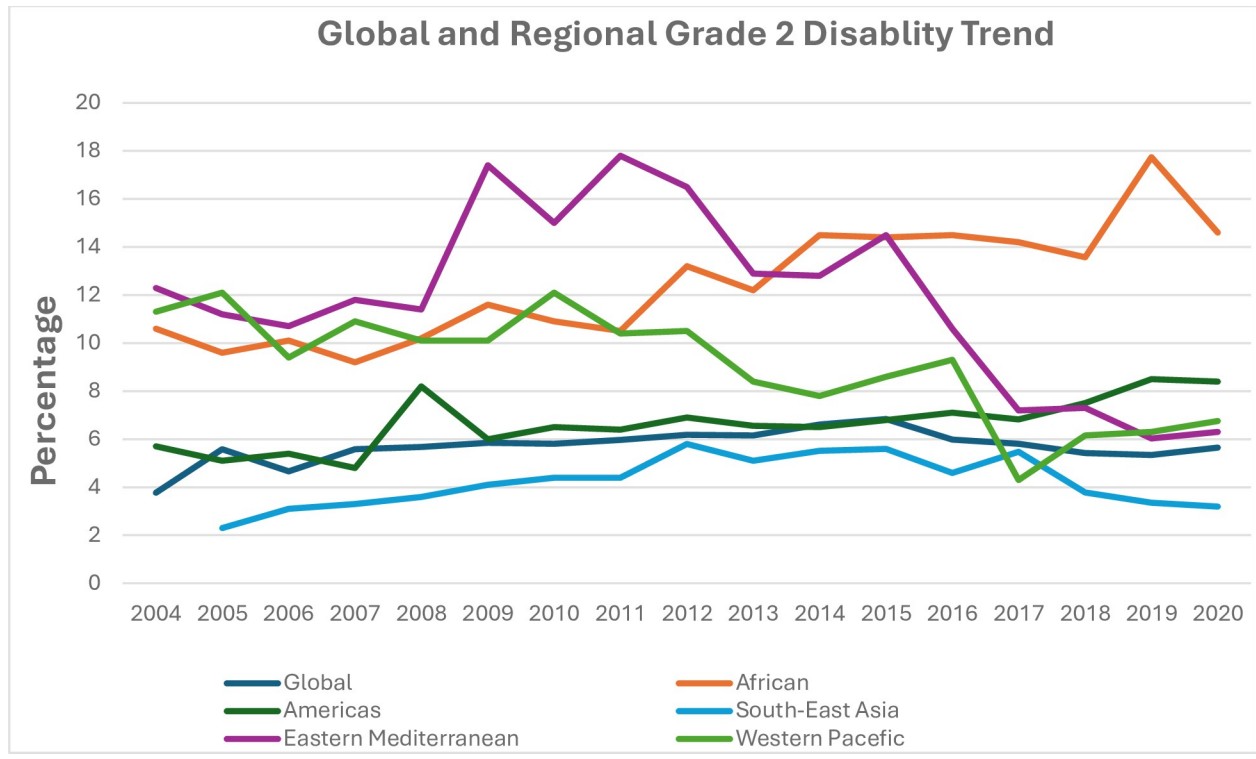

**Fig 2. The global and regional percentage of G2D among newly diagnosed leprosy patients from 2004 to 2020.**

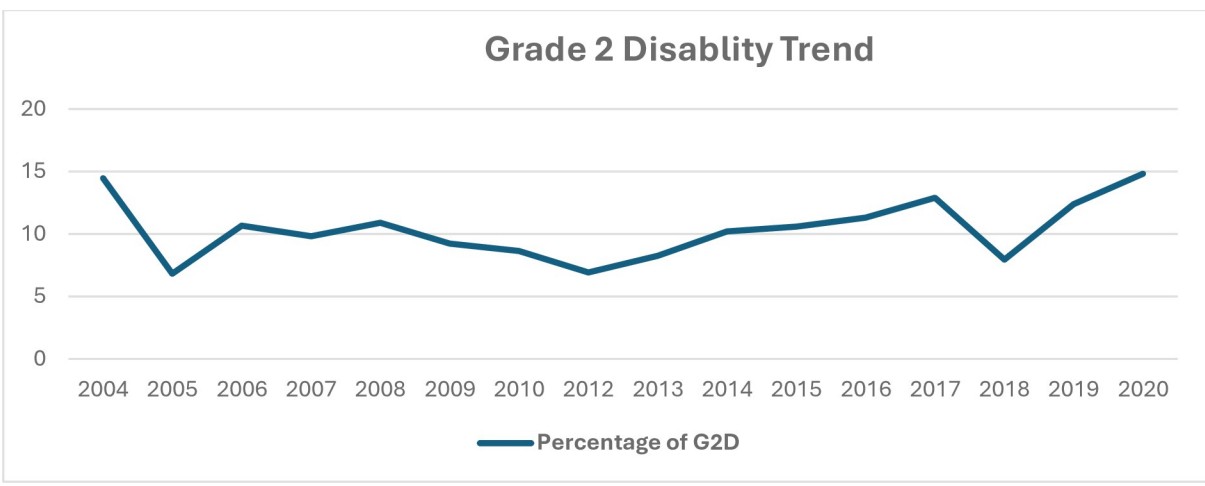

**Fig 3.  The percentage of G2D among newly diagnosed leprosy patients from 2004 to 2020 in Ethiopia.**

### Review of leprosy patients diagnosed at ALERT hospital (January 2016 to December 2020)

Totally 1032 leprosy patients were seen at ALERT Hospital from 1st January 2016 to 31st December 2020.Out of these 660 patients were referred to other clinics to complete MDT. There were 309 patients who completed MDT at ALERT. The remaining 63 cases were registered on the leprosy patient registry book but discontinued medication after they received one dose of MDT or taking it for few months making the dropout rate to be 16.9%.

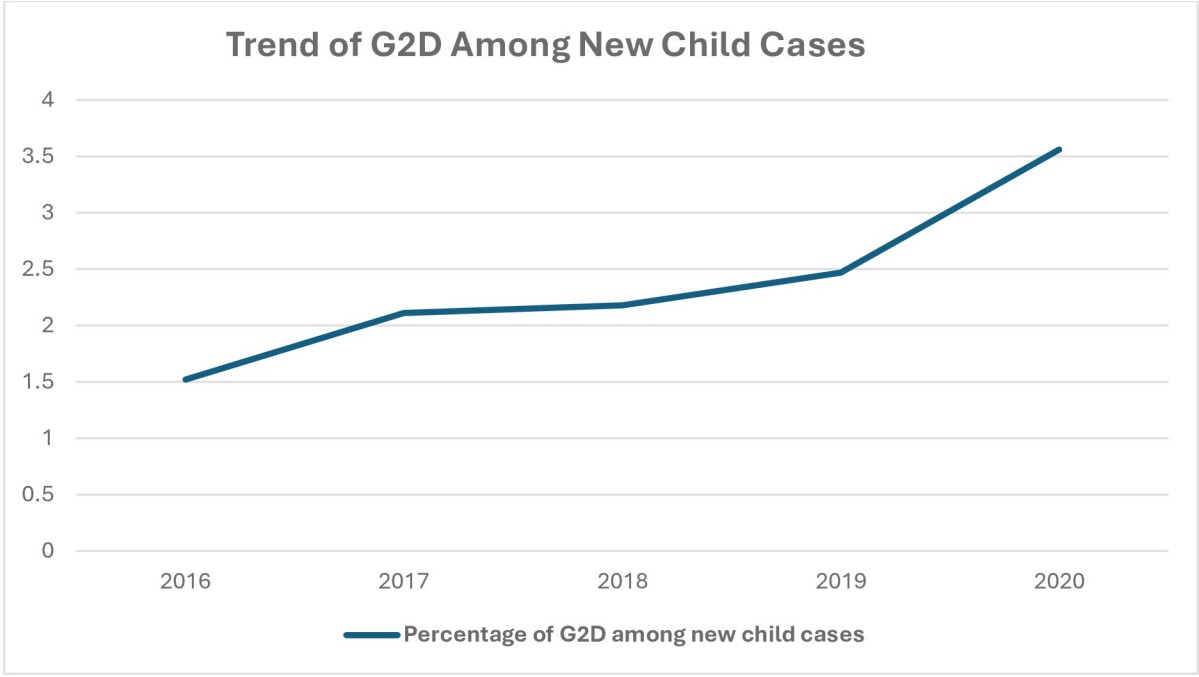

**Fig 4.  The global percentage of G2D among newly diagnosed child cases from 2016 to 2020.**

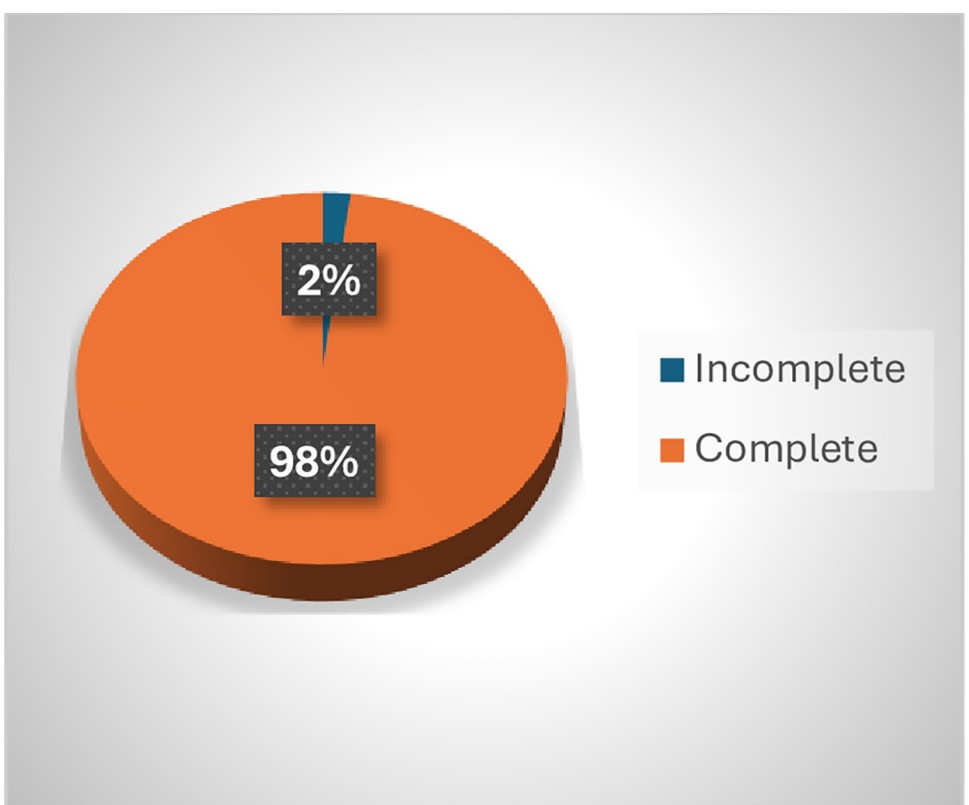

**Fig 5. Disability data at Diagnosis.**

Disability data at diagnosis was complete for 303 (98%) of cases while 197(64.8%) of them had complete data at the end of MDT (**Figs 5 and 6**). The baseline sociodemographic variables (sex, age, region, year of diagnosis) were almost filled completely while the other components regarding their household contact were not filled at all.

Among the total number of 309 patients who started and completed MDT at ALERT hospital 202(65.37%) were males and 107(34.6%) were females. The median age was 30 years where children (<15 years) were 29(9.4%). One hundred seventy-nine (58.0%) were from Addis Ababa and majority of the remaining cases were from Oromia and Amhara region (**Table 1**). Out of 660 patients who had been transferred to the regional health facilities 470 (71.2%) were male and 190 (28.9%) females. The median age was 36. Among those patients transferred out 229(36.5%) had G1D and 247(39.4%) of them had G2D at diagnosis. Most of the cases were transferred to Oromia and Amhara region (**Table 1**).

Out of the 63 leprosy patients who had discontinued treatment, nearly three fourth of them were male and half of them were older than 30 years of age. More than ninety percent of them were multibacillary. G0D and G1D cases were 14(23.3%) while G2D cases were 32(53.33%) (**Table 2**).

Among the 196 patients whose disability grade was recorded at completion of MDT,55 (28.1%) showed improvement on their disability grade,72(36.7%) remained the same and 7 (3.6%) deteriorated in their disability grade. From 29 children (<15 years) with leprosy grade recorded at diagnosis 15(51.7%) had G0D, 6(20.7%) had G1D and 8(27.6%) had G2D while out of the 14 children with disability grade recorded at completion 3(35.6%) had G0D, 3 (21.5%) had G1D and 6(42.9%) had G2D (**Table 2**).

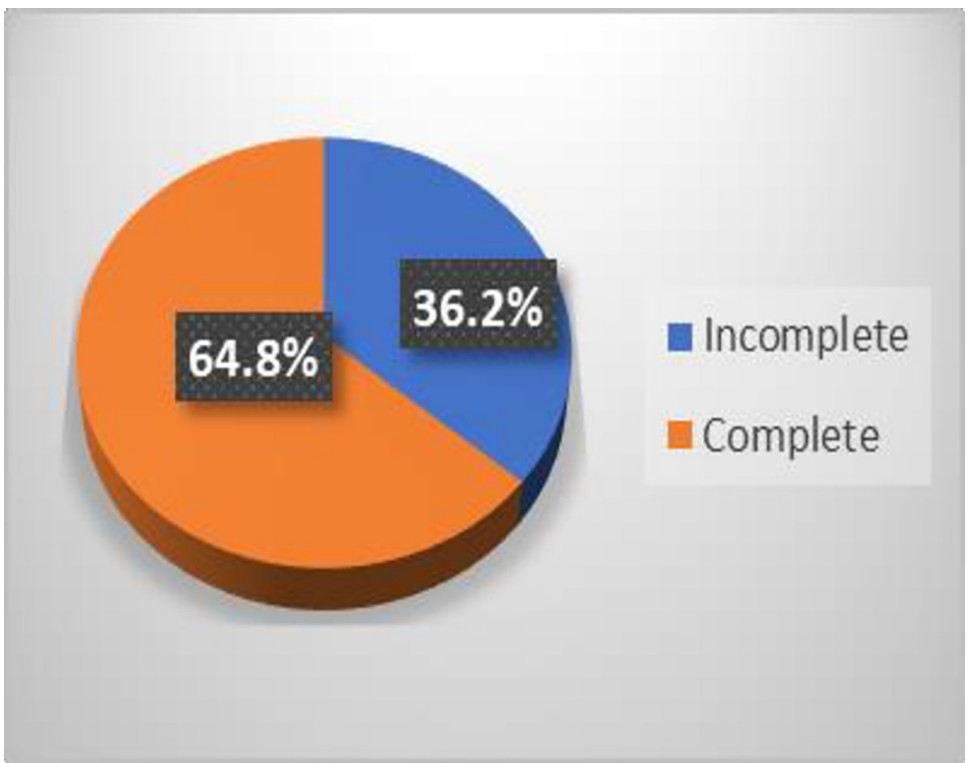

**Fig 6. Disability data at completion MDT.**

Out of the 303 patients for whom WHO disability grade was recorded at diagnosis 101 (51.5%) were Grade 0, 50 (25.5%) were Grade 1 and 45(22.9%) of them were Grade 2(**Table 3**).

## Change in disability grade at completion of MDT

**Table 4** shows factors associated with the presence of disability (either Grade 1 or 2) at baseline. A univariate logistic regression showed age greater than 30 was associated with increased odds of disability presence and this was statistically significant (OR 1.94, 95% CI 1.18–3.17). Multivariate analysis showed a consistently increased odds of disability presence with age after adjusted for gender, leprosy category, smear result, and bacteriologic index, with an AOR 2.9 (1.02–8.32) and p-vale of 0.008.

## Interview with health care workers (HCWs) and Professionals working in Leprosy

All of the five healthcare workers based in the leprosy clinics at ALERT, agreed with the gap in appropriately recording and reporting disability data attributing it to forgetfulness and lack of emphasis. Furthermore, they believe that there is different level of understanding about the significance of disability data recording among health professionals working in leprosy clinics. Two of the healthcare workers mentioned difficulty of accepting their disease condition by patients considering leprosy as a genetic disease and absence of family history is one of the reason most patients discontinue medication and follow-up while another health worker pointed out that patients also discontinue medications when they develop leprosy reactions relating it to medication side effect.

**Table 1. Socio-demographic characteristics of leprosy patients diagnosed at ALERT referral hospital Addis Ababa, Ethiopia (September 2016-December 2020).**

| Variables | Completed MDT Total (N = 309) | Transfer Out Total (N = 660) | Discontinued MDT Total (N = 63) |
|---|---|---|---|
| Sex | | | |
| Male | 202(65.4) | 470(71.2) | 47(74.5) |
| Female | 107(34.6) | 190(28.9) | 16 (25.3) |
| Age years | | | |
| Mean (±SD) | 33.6(±13.8) | 38.8(±16.7) | 35.8(±12.9) |
| Median | 30 | 36 | 34 |
| Min–Max | 3–83 | 2–92 | 5–90 |
| Age group | | n = 660 | |
| 1–10 | 2(0.7) | 8(1.2) | 2(3.2) |
| 11–20 | 41 (13.3) | 92(13.9) | 5(7.9) |
| 21–30 | 122(39.5) | 166(25.2) | 28(44.4) |
| 31–40 | 71(23.0) | 138(20.9) | 18(28.6) |
| 41–50 | 35(11.4) | 83(12.6) | 5(7.9) |
| 51–60 | 24(7.8) | 104(15.8) | 3(4.8) |
| 71–80 | 3()1.0 | 11(1.7) | 1(1.6) |
| 81–90 | 1(0.3) | 3(0.5) | 0(0.0) |
| 90–100 | 0(0.0) | 1(0.2) | 0(0.0) |
| Year of Diagnosis | | | |
| 2016 | 37(12.0) | 97(14.7) | 12(19.0) |
| 2017 | 64(20.7) | 178(27.0) | 9(14.3) |
| 2018 | 64(20.7) | 141(21.4) | 15(23.8) |
| 2019 | 93(30.1) | 121(18.3) | 14(22.2) |
| 2020 | 51(16.5) | 123(18.7) | 13(20.6) |
| Region | | | |
| Addis Ababa | 179(58.0) | 10(1.5) | 1(1.6) |
| Dire Dawa | 0(0.0) | 0(0.0) | 0(0.0) |
| Tigray | 1(0.3) | 1(0.2) | 2(3.2) |
| Amhara | 36(11.7) | **213(32.3)** | 22(34.9) |
| Oromia | 83(26.9) | **361(54.7)** | 30(47.6) |
| Harari | 0(0.00) | 7(1.0) | 5(7.9) |
| Somali | 0(0.00) | 0(0.0) | 0(0.0) |
| Afar | 1(0.3) | 7(1.1) | 0(0.0) |
| SNNPRs | 9(2.9) | 4(0.6) | 2(3.2) |
| Gambella | 0(0.0) | 64(9.7) | 1(1.6) |
| Benishangul-Gumuz | 0(0.0) | 0(0.0) | 0(0.0) |

*Data are in the order of numbers(N) followed by percentage

Three of the professionals working in leprosy mentioned that they have observed lack of knowledge and skill in properly assessing disability in some clinics during their supervision and this is even more profound in rural health facilities when data verification is done. This is further augmented by high turnover rate and shortage of trained health care professionals working in leprosy clinics.

Four of the leprosy professionals believe recording disability data at completion of MDT is important as it helps to understand the effectiveness of leprosy control programmes at national level and help the WHO to know the disability burden more precisely adding to the existing

**Table 2. Clinical characteristics of leprosy patients diagnosed at ALERT referral hospital Addis Ababa, Ethiopia (September 2016-December 2020).**

| Variables | Completed MDT (N = 309) | Transfer Out (N = 660) | Discontinued MDT (N = 63) |
|---|---|---|---|
| Type of Leprosy (WHO Classification) | | | |
| PB | 26(8.5) | 73 (11.1) | 4(6.1) |
| MB | 279(91.5) | 583(88.9) | 59(93.9) |
| Smear Result | n = 305 | n = 660 | n = 55 |
| Positive | 160(52.5) | 331(50.1) | 30(54.5) |
| Negative | 145(47.5) | 329(49.9) | 25(45.5) |
| Duration of Treatment (months) | n = 303 | | |
| Mean PB MB | 13.3 15.2 | - - | - - |
| Disability grade at diagnosis | n = 303 | n = 627 | n = 60 |
| Grade 0 | 98(32.3) | 151(24.1) | 14(23.3) |
| Grade 1 | 105(34.7) | 229(36.5) | 14(23.3) |
| Grade 2 | 100(33.0) | 247(39.4) | 32(53.3) |
| Disability grade at completion | n = 196 | | |
| Grade 0 | 101(51.5) | - | - |
| Grade 1 | 50 (25.5) | - | - |
| Grade 2 | 45 (23.0) | - | - |
| Change in Disability Grade | n = 196 | | |
| Improved | 55(28.0) | - | - |
| Same | 134(68.4) | - | - |
| Deteriorated | 7(3.6%) | - | |
| Corticosteroid Use | n = 286 | | |
| Yes | 96 (33.6) | - | - |
| No | 190 (66.4) | - | - |
| Reason for steroid Use | n = 96 | | |
| Type 1 Reaction | 56(58.3) | - | - |
| ENL | 36(37.5) | - | - |
| Neuritis Disability in Children at diagnosis (<15 years) Grade 0 Grade 1 Grade 2 | 4(4.2) n = 29 5(51.7) 6(20.7) 8(27.6) | - - - - | - - - - |
| Disability in Children at completion (<15 years) | n = 14 | | |
| Grade 0 Grade 1 Grade 2 | 5(35.6) 3(21.5) 6(42.9) | | |

*Data are number (%) unless otherwise specified

data of persons detected with disability at diagnosis in different countries and thus help in improving quality of treatment and reducing disability worsening while on treatment.

Two of the professionals working in leprosy agree that G1D should also be reported to WHO despite the current practice of reporting only G2D as it helps to design strategies and in preventing further deterioration in disability level. Despite its significance, one of the professionals pointed out that WHO is not requesting for G1D data as assessing for anaesthesia is skill requiring and difficult for most frontline health workers as there is lack of standard

**Table 3. Disability grade at diagnosis of leprosy patients who completed MDT at ALERT referral hospital Addis Ababa, Ethiopia (January 2016-December 2020). N = 303.**

| Variables | Grade 0 (%) | Grade 1(%) | Grade 2(%) | Total (%) |
|---|---|---|---|---|
| | n = 98 | n = 105 | n = 100 | N = 303 |
| Sex | | | | |
| Male | 62(63.3) | 68(64.7) | 67(67.0) | 197(65.0) |
| Female | 36(36.7) | 37(35.2) | 33(33.0) | 106(35.0) |
| Age | | | | |
| 1–10 | 0(0.0) | 0(0.0) | 2(2.0) | 2(0.7) |
| 11–20 | 19(19.4) | 7(6.7) | 15(15.0) | 4(1.3) |
| 21–30 | 45(45.9) | 42(40.0) | 33(33.0) | 119(39.3) |
| 31–40 | 18(18.4) | 27(14.2) | 23(23.0) | 68(22.4) |
| 41–50 | 11(11.2) | 15(14.3) | 9(9.0) | 35(11.5) |
| 51–60 | 3(3.1) | 8(7.6) | 13(13.0) | 24(7.9) |
| 61–70 | 2(2.0) | 4(3.8) | 3(3.0) | 9(3.0) |
| 71–80 | 0(0.0) | 2(1.9) | 1(1.8) | 3(1.0) |
| 81–90 | 0(0.0) | 0(0.0) | 1(0.3) | 1(0.3) |
| Year of Diagnosis | | | | |
| 2016 | 11(11.2) | 8(7.6) | 17(17.0) | 36(11.9) |
| 2017 | 24(24.5) | 15(14.3) | 25(25.0) | 64(21.1) |
| 2018 | 26(26.5) | 19(18.1) | 14(14.0) | 59(19.5) |
| 2019 | 21(21.4) | 40(38.1) | 33(33.0) | 94(31.0) |
| 2020 | 16(18.3) | 23(21.9) | 11(11.0) | 50(16.5) |
| Bacteriologic Index | | | | |
| 0 | 44(44.9) | 54(51.4) | 58(58.0) | 156(51.4) |
| 1+ | 7(7.1) | 10(9.5) | 4(4.0) | 21(6.9) |
| 2+ | 11(11.2) | 10(9.5) | 9(9.0) | 30(9.9) |
| 3+ | 13(13.3) | 13(12.4) | 9(9.0) | 35(11.5) |
| 4+ | 8(8.2) | 6(5.7) | 9(9.0) | 23(7.6) |
| 5+ | 11(11.2) | 9(8.6) | 4(4.0) | 24(7.9) |
| 6+ | 4(4.1) | 3(2.8) | 7(7.0) | 14(4.6) |
| Category | n = 95 | n = 104 | n = 99 | N = 298 |
| PB | 10(10.5) | 7(6.7) | 9(9.1) | 26(8.7) |
| MB | 85(89.5) | 97(93.2) | 90(90.9) | 272(91.3) |
| Smear Result | n = 95 | n = 103 | n = 100 | N = 298 |
| Positive | 56(58.9) | 56(54.4) | 44(44.0) | 156(52.3) |
| Negative | 39(41.1) | 47(45.6) | 56(56.0) | 142(47.7) |

*Data are number (%) unless otherwise specified

definitions. Furthermore, two of the professionals believe that new sets of standard operating procedures for WHO Leprosy disability grading is required tailored in a way suitable for individual case management as it was initially designed for public health programme.

## Discussion

The analysis done from the WER of WHO conveys globally the number and prevalence of G2D among newly diagnosed leprosy cases is generally stable where the prevalence is between 4 and 6 percent with minimum fluctuations for the past 20 years. Despite significant variation between the regions of the world, the trend of G2D has been persistently increasing among

**Table 4. Factors associated with presence of disability in leprosy patients who were diagnosed at ALERT referral hospital Addis Ababa, Ethiopia (September 2016-December 2020).**

| Variables | DISABLITY | | COR | P-Value | 95% CI | | AOR (95% CI) |
|---|---|---|---|---|---|---|---|
| | **Absent** | **Present** | | | | | |
| Gender | | | | | | | |
| Male | 61(62.9%) | 136(65.9%) | 1 | | | | |
| Female | 36(37.1%) | 70(34.1%) | 0.88 | 0.614 | (0.53–1.45) | | |
| Age | | | | | | | |
| <30 years | 63(64.3%) | 99(48.3%) | 1 | | | | 1 |
| >30years | 35(35.7%) | 106(52.7%) | **1.94** | **0.008** | **(1.18–3.17)** | | **2.90(1.02–8.32)** |
| Category | | | | | | | |
| PB | 10(10.5%) | 16(7.9%) | 1 | | | | |
| MB | 85(89.5%) | 187(92.1%) | 1.43 | 0.402 | (0.62–3.27) | | |
| Smear Result | | | | | | | |
| Positive | 56(58.9%) | 100(49.2%) | | | | | |
| Negative | 39(41.1%) | 103(50.8%) | 1.46 | 0.128 | (0.89–2.39) | | |
| Bacteriologic Index | | | | | | | |
| BI <4+ | 31(31.6) | 58(27.3) | 1 | | | | |
| BI ≥4+ | 67(68.4) | 154(72.6) | 1.2 | 0.44 | (0.72–2.07) | | |

new cases in African region and relatively have stable pattern with no decline in the Southeast Asian and American regions where most of the leprosy patients are found. Furthermore, the prevalence of G2D among new cases has been increasing in Ethiopia since 2012 (**Fig 3**) and similar trend is observed at ALERT indicating that the trend of G2D increasing in high endemic regions of the world. This strongly shows leprosy patients are still being diagnosed late to the level they have already developed the highest disability grade at diagnosis. This in turn clearly depicts the gap in early identification and proper management of leprosy patients and prevention of transmission. Furthermore, the decline of reported leprosy cases and disability in some regions of the world in 2020 could be to the effect of COVID 19 pandemic on leprosy programmes as there was a 37% reduction in detection of leprosy cases [9].

With the observed increasing trend of G2D, it would be difficult to achieve the goals set by WHO roadmap of NTDs by 2030 which targets to reduce the annual number of new leprosy cases below 62,500; rate (per million population) of new cases with grade 2 disability to be below 0.12 and rate (per million children) of new paediatric cases with leprosy to be under 0.77 [16].

Furthermore, the study done in ALERT gave an opportunity to closely observe leprosy associated disability in a typical leprosy endemic country like Ethiopia. Among all patients diagnosed at ALERT hospital nearly two-third of them were transferred back to their nearest health facility to continue treatment indicating most cases are coming from outside Addis Ababa (**Table 1**).This could be due to leprosy cases are still referred to ALERT for confirmation as HCWs at peripheral centre may lack the confidence to make a definite diagnosis; less awareness about the disease in the community to seek medical care thus presenting at late stage with disability requiring referral or people may prefer to be seen at referral centres assuming they would get better care. Disability data at the end of MDT was only 64.8% complete among those who were treated at ALERT referral hospital. More than half of both G1D and G2D was recorded in the age group between 21- and 40-years depicting disability is more pronounced in the younger and productive group of population showing the impact of leprosy in worsening poverty in endemic countries. Among those leprosy patients who completed MDT at ALERT hospital 29 (9.4%) of them were children (<15 years) (**Table 2**).This figure is

lower compared to other leprosy endemic African countries like Comoros where it was 27% in 2016 but if further active case surveillance was implemented the number could get even higher. But it is still higher compared to other countries Kenya where it was 7.5% in 2014 and Benin where it was 2.6% in 2018 [17,18]

The ratio of male to female patients who started and completed MDT at ALERT was 1.9 (**Table 3**).This male predominance was also consistent with other studies done in China, Nepal, Brazil and Nigeria [16,19].Among those patients who completed treatment at ALERT with G1D and G2D,64.7% and 67% of them were male respectively This indicates male are almost two times more likely to develop disability compared to females and a metanalytic study done in 2019 has also showed strong association between male sex and disability [20]. This could be due to the hesitance and negligence of men in seeking health care services early as they may ignore the initial symptoms and later present with advanced health problem. It is also higher than global and national target of childhood leprosy among new cases to be below 5% [21].This marks an active transmission of leprosy in the community. This study has found that more than ninety percent of patients are MB cases, and this high prevalence of MB cases is consistent with other studies done in China and western pacific region [22, 23].Smear positivity is seen in more than 50% of patients diagnosed at ALERT and this could be due to late diagnosis and poses significant risk of transmission in the community and alarms high index of suspicion.

The high proportion of G2D at diagnosis seen in ALERT could be due to delayed diagnosis before presenting to referral centre or could be due to referral of complicated cases. More than half of patients who have discontinued treatment had G2D at diagnosis and this could be because of stigma associated with the visible deformities hindering continuous visit to health facilities; due to socioeconomic reasons due to disability as disabled people may not work enough and earn money to cover expenses for hospital visit or patients could give up with treatment after some irreversible damages has happened. Also, it could be related with the quality of services provided at ALERT.

The disability grade at completion of MDT at ALERT showed that most patients' initial recorded disability grade improved or stayed the same and only 3.6% of cases showed worsening of disability grade while on treatment. But these results differ from studies done in Indonesia where 15.8% of cases showed disability worsening at completion of MDT and has similar percentage of unrecorded data (37.7%) [10]. Another survival study done in Brazil which took patients with complete disability recording reported no significant change in disability status at the end of treatment [24]. This low occurrence of disability worsening at ALERT could be due to the fact that only 61.7% of patients and 48.3% of children disability status was evaluated at the end of treatment and this may have underestimated the actual figure of prevalence of disability worsening. Moreover, problems with proper disability assessment and data recording may have contributed.

In this study further analysis for risk factors associated with disability has found that being older (age>30) is a risk factor for developing disability (**Table 4**). Previously done study in ALERT hospital in 2013 and another study from China has also showed the risk of disability increases with age [13,25]. The higher prevalence of disability in older age group in ALERT could be due to delayed diagnosis of patients because of absence of early case identification in peripheral health facilities and lack of awareness about the early signs of the disease in the community.

The findings from ALERT hospital have also shown that disability status is not static subject to changes in course of time and could even worsen during treatment. This holds true in the post MDT period where findings of study done in Brazil and Indonesia has demonstrated worsening of disability after patients are released from treatment centre [10,24]. Other study

from Brazil predicted the probability of disability progression to be 35% after 15 years post MDT [24]. Therefore, collecting disability data by WHO at diagnosis (baseline) only may not be sufficient enough in estimating the actual disability burden in persons affected by leprosy. It is also useful to assess the effectiveness of leprosy treatment services given at health facilities in preventing disability while patients are on treatment especially related to managing leprosy reactions and drug adherence.

Interview with health care workers and leprosy professionals has also brought to light the limitations in reliability and accuracy of disability data reported to the WHO compromising public health decisions made towards leprosy. Most of the participants interviewed believe there is a problem in disability data quality and completeness which is noted to be poorer in regional health facilities. This could be related to lack of adequate knowledge and skills among health workers in performing proper disability grade assessment or recording data. A study done in selected health facilities in Oromia and Amhara regions in Ethiopia aimed at assessing the knowledge and skills of healthcare workers engaged in management of leprosy in public health facilities showed that it was unsatisfactory as only 18% of respondents diagnosed leprosy correctly and 86% had poor knowledge [26].

The leprosy experts have pointed out limitations of operational definitions laid out by WHO to be used by frontline health workers indicating they are subject to different interpretations among various programmes. For example, Grade 2 disability is generally defined as 'Visible deformity or damage present' by WHO and scars are usually graded as Grade 2 by health professionals as they are healed ulcers and visible damage that can be seen easily [27]. Despite this, operational and expanded definitions excludes scars from being counted as Grade 2 showing disability has not been counted appropriately and uniformly across programmes [11].

The way WHO counts disability was also criticized by the leprosy experts as it only gets report of G2D disability data at diagnosis not taking into consideration to the disability change that occurs post diagnosis. This is also evident from the finding of this study in ALERT hospital which showed 6.15% of patients with G1D disability at diagnosis has deteriorated to G2D at completion of MDT(**Figs 7 and 8**). But since WHO counts disability at diagnosis only, it is missing significant proportion of patients with G2D,thus lacking accurate information on the actual burden of G2D(**Fig 9**). Therefore, collecting disability data at the end of MDT by WHO is crucial to get relatively better estimate of the actual burden of G2D worldwide.

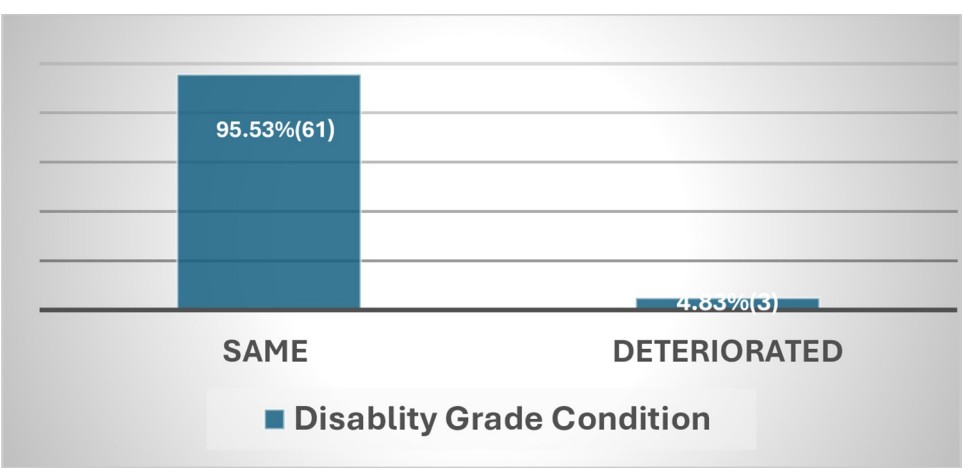

**Fig 7. Patients with grade 0 disability at baseline.**

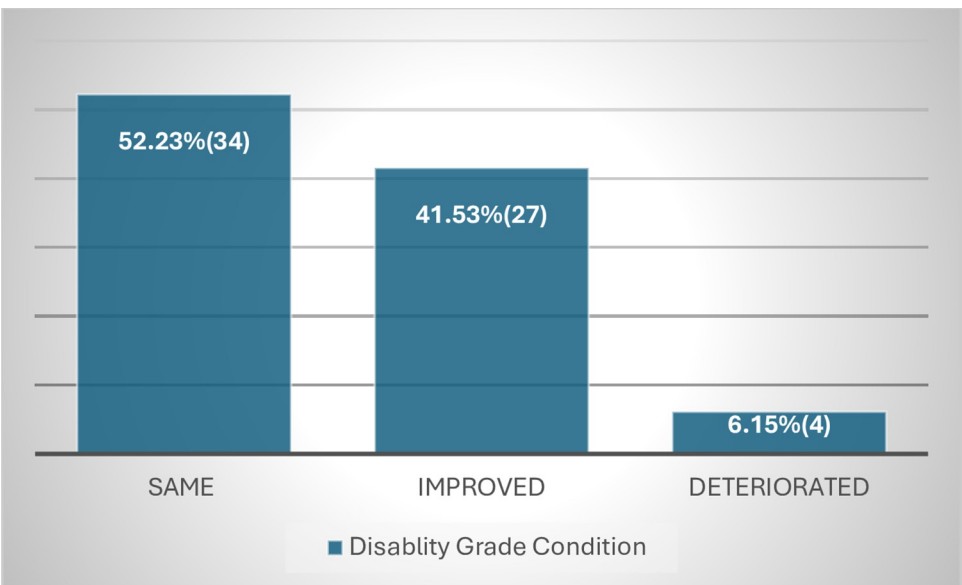

**Fig 8. Patients with grade 1 disability at baseline.**

Furthermore, the leprosy professionals suggested collecting G1D data at diagnosis will give WHO the capacity to effectively monitor and evaluate national leprosy programs towards management and prevention of disability progression (*Fig 10*). As the current strategies focuses on integrated and comprehensive case management of patients, knowing the burden of G1D would also be crucial for global and national leprosy programmes to design strategies that could target in prevention of disability progression and deliver better services focusing on individual case management [28].As it was mentioned by one of the leprosy professionals, the challenge of assessing G1D is the higher level skill it requires making it difficult for the

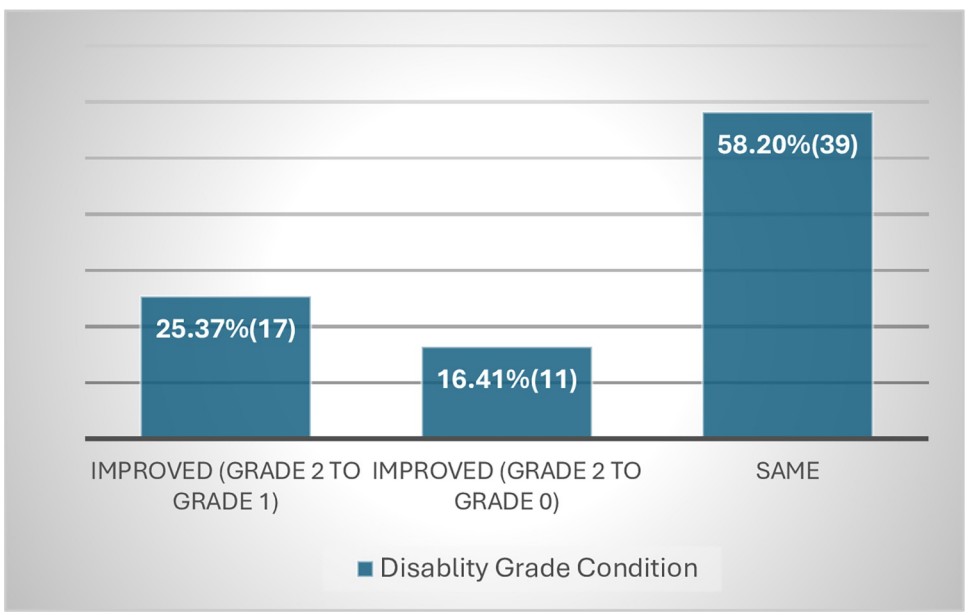

**Fig 9. Patients with grade 2 disability at baseline.**

| HCWs in Leprosy Clinics at ALERT | Leprosy Professionals at national and global level |
|---|---|
| ■ All the HCWs believe that there is gap in leprosy disability data recording and reporting<br><br>■ All of them believe there is variation of understanding among HCWs about the significance of recording and reporting disability data<br><br>■ Two of the HCWs mentioned denial of illness and leprosy reactions as a reason for medication | ■ Three of them mentioned observing knowledge and skill gap in assessing disability during their supervision<br>■ Four of them agree in the usefulness of disability data recording completion of MDT<br>■ Two of them agreed that G1D should be reported to WHO<br>■ Two of them believe that WHO should come up with a new standard operating definition for disability grading |

**Fig 10. Summary of interview with health care workers (HCWs) and Professionals working in Leprosy.**

frontline health workers indicating the need for better and easier point of care test for assessing nerve function.

One of the limitations of this study is the collection of data only from tertiary level hospital. Since ALERT hospital is a referral hospital, most of the cases seen are likely to have severe disease condition. This could have overestimated the prevalence of disability most importantly G2D.The other limitation is related with data reliability as this is a retrospective study there could be issues with the accuracy of the data recorded. However, the findings of the study are vital in understanding about the condition of disability grade at the end of MDT and its significance based on the data from ALERT hospital and showed the gaps in WHO disability data.

## Conclusions

The analysis of trend of G2D showed it remains stable globally not showing any significant decline for the past 20 years but increasing in the African region and high in endemic countries like Ethiopia.

The review of leprosy patients diagnosed at ALERT hospital data has shown the prevalence of disability at diagnosis (both G1D and G2D) is still high among newly diagnosed leprosy patient indicating a problem in early case identification and prompt treatment. There is also decline of disability data at the completion of MDT.

The study from ALERT and the interview with HCWs and leprosy professionals has shown the limitations of the data reported to WHO in terms of completeness and quality due problems in accurate disability assessment and less emphasis given for proper data recording by HCWs in leprosy clinics suggesting WHO may not be collecting reliable data.

The way WHO counts disability and the actual definitions used by different programmes and health workers varies and are prone to interpretation bias thus WHO may not be counting and getting the true burden of disability in endemic countries. G1D data could also be used as an important parameter for individual case management. Since disability grade condition is not static and changes in course of time, prospective studies should be done to assess disability progression post MDT and come up with possible solutions in preventing them.

Finally, the findings of this study indicate there are still significant challenges and setbacks towards attaining the leprosy targets outlined by who to be achieved by 2030.

## Recommendation

Based on the findings of this study and in relation to the global targets set by WHO for leprosy, it is recommended that an integrated approach in prevention and management of leprosy associated disability should be followed.

Frontline healthcare workers should be well trained to identify leprosy cases early and those working in leprosy clinics should be well equipped with the necessary knowledge and skill to manage leprosy reactions, patient counselling, accurate disability grading and reporting.

Further emphasis should be given for the quality of disability data reported at national and global level through data verification and continuous feedback.

Collecting disability data at completion and post MDT is recommended as this will help national leprosy programs and WHO to know the burden of disability more precisely and to further monitor and evaluate the effectiveness of their programs.

The issue of reporting G1D should be further studied in terms of feasibility and getting reliable data as knowing the prevalence of G1D is crucial on designing individual cased based approaches focusing on prevention of disability progression.

WHO should come up with leprosy disability grading system which can be uniformly applied across programmes and use parameters which can be used to capture the actual disability burden that can be used to guide better individual case management and prevention of disability for the successful achievement of targets set by WHO to be achieved by 2030.

## Supporting information

**S1 Data. Data of patients who had completed multi-drug therapy at ALERT Hospital (September 2016-December 2020).**
(XLS)

**S2 Data. Data of patients who were transferred from ALERT hospital to other health facilities for completion of multi-drug therapy (September 2016-December 2020).**
(XLSX)

## Author Contributions

**Conceptualization:** Bereket Abebayehu Tegene.

**Data curation:** Bereket Abebayehu Tegene, Thomas Asfaw Atnafu.

**Formal analysis:** Bereket Abebayehu Tegene.

**Investigation:** Bereket Abebayehu Tegene.

**Methodology:** Bereket Abebayehu Tegene.

**Project administration:** Bereket Abebayehu Tegene, Thomas Asfaw Atnafu.

**Resources:** Bereket Abebayehu Tegene.

**Software:** Bereket Abebayehu Tegene.

**Supervision:** Bereket Abebayehu Tegene.

**Validation:** Bereket Abebayehu Tegene, Thomas Asfaw Atnafu.

**Visualization:** Bereket Abebayehu Tegene, Thomas Asfaw Atnafu.

**Writing – original draft:** Bereket Abebayehu Tegene.

**Writing – review & editing:** Bereket Abebayehu Tegene.

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
