## [Decision Letter · Decision Letter 0]

13 Aug 2024

Dear Dr. Tegene,

Thank you very much for submitting your manuscript "Relevance of reporting leprosy related disability at the completion of multi drug therapy: A 5-year retrospective analysis of disability in persons affected by leprosy at ALERT Hospital Ethiopia" for consideration at PLOS Neglected Tropical Diseases. As with all papers reviewed by the journal, your manuscript was reviewed by members of the editorial board and by several independent reviewers. In light of the reviews (below this email), we would like to invite the resubmission of a significantly-revised version that takes into account the reviewers' comments. 

It is an important work showing the trend of Grade 2 Disability (G2D) among newly diagnosed leprosy patients shows no decline globally over the last 20 years. It shows that, in fact, it is increasing in Africa. At ALERT Hospital, the G2D rate from 2016 to 2020 was 33%, that is very high.

Reference: The references need extensive review.

Figure 1: Figure 1 needs revision; it was cut off.

Disability Data: The global and WHO regional disability data were presented only as percentages. It is important to also present these data in absolute numbers. Additionally, the authors should discuss the current number of people with disabilities due to leprosy worldwide. If the absolute numbers are decreasing, potential reasons should be explored, including the impact of the COVID-19 pandemic and the poor awareness mentioned in the discussion. The WHO estimates that between 2 and 3 million people are currently living with disabilities due to leprosy, and this should also be discussed.

Line 206: Please include the dropout rate at ALERT Hospital. It appears that 309 patients were treated there, and 63 received the first dose but did not return. What would the dropout rate be in this case? This is important because more than half of these patients had G2D.

Line 215: “Error! Reference source not found.”

Table 2: More than 50% of the diagnosed patients were skin smear positive. This is very high. Could this be due to late diagnosis? Please discuss.

Evaluation of Disability Grades: People evaluated for disability grade at diagnosis: 303; people evaluated for disability grade at completion: 196 (64.7%). The situation is worse in children: 29 to 14 (48.3%). The change in disability may be much worse than depicted. This needs to be part of the discussion starting at line 327, where the low percentage of disability worsening at ALERT is correlated with the high quality of care at ALERT. However, it should be noted, as mentioned in the previous paragraph, that more than half of the patients with G2D discontinued treatment. High-quality care should prevent this. Other possibilities should be discussed.

Line 310: Could you please further discuss the issue of 9.4% of children among those who completed treatment at ALERT? Although 9.4% seems high, in other countries, like the Comoros or some regions of Indonesia, these percentages can exceed 30%. If an active search is implemented, perhaps the percentage of children diagnosed with leprosy would increase. Regarding the other African countries mentioned, is the lower percentage a good or bad indicator of leprosy diagnosis in those countries?

Repetitive Citations: It is not necessary to repeat the citation of tables and/or figures during the discussion.

We cannot make any decision about publication until we have seen the revised manuscript and your response to the reviewers' comments. Your revised manuscript is also likely to be sent to reviewers for further evaluation.

Sincerely,

Claudio Guedes Salgado, PhD

Academic Editor

Mathieu Picardeau

Section Editor

It is an important work showing the trend of Grade 2 Disability (G2D) among newly diagnosed leprosy patients shows no decline globally over the last 20 years. It shows that, in fact, it is increasing in Africa. At ALERT Hospital, the G2D rate from 2016 to 2020 was 33%, that is very high.

Reference: The references need extensive review.

Figure 1: Figure 1 needs revision; it was cut off.

Disability Data: The global and WHO regional disability data were presented only as percentages. It is important to also present these data in absolute numbers. Additionally, the authors should discuss the current number of people with disabilities due to leprosy worldwide. If the absolute numbers are decreasing, potential reasons should be explored, including the impact of the COVID-19 pandemic and the poor awareness mentioned in the discussion. The WHO estimates that between 2 and 3 million people are currently living with disabilities due to leprosy, and this should also be discussed.

Line 206: Please include the dropout rate at ALERT Hospital. It appears that 309 patients were treated there, and 63 received the first dose but did not return. What would the dropout rate be in this case? This is important because more than half of these patients had G2D.

Line 215: “Error! Reference source not found.”

Table 2: More than 50% of the diagnosed patients were skin smear positive. This is very high. Could this be due to late diagnosis? Please discuss.

Evaluation of Disability Grades: People evaluated for disability grade at diagnosis: 303; people evaluated for disability grade at completion: 196 (64.7%). The situation is worse in children: 29 to 14 (48.3%). The change in disability may be much worse than depicted. This needs to be part of the discussion starting at line 327, where the low percentage of disability worsening at ALERT is correlated with the high quality of care at ALERT. However, it should be noted, as mentioned in the previous paragraph, that more than half of the patients with G2D discontinued treatment. High-quality care should prevent this. Other possibilities should be discussed.

Line 310: Could you please further discuss the issue of 9.4% of children among those who completed treatment at ALERT? Although 9.4% seems high, in other countries, like the Comoros or some regions of Indonesia, these percentages can exceed 30%. If an active search is implemented, perhaps the percentage of children diagnosed with leprosy would increase. Regarding the other African countries mentioned, is the lower percentage a good or bad indicator of leprosy diagnosis in those countries?

Repetitive Citations: It is not necessary to repeat the citation of tables and/or figures during the discussion.

Reviewer's Responses to Questions

**Key Review Criteria Required for Acceptance?**

**Methods**

-Are the objectives of the study clearly articulated with a clear testable hypothesis stated?

-Is the study design appropriate to address the stated objectives?

-Is the population clearly described and appropriate for the hypothesis being tested?

-Is the sample size sufficient to ensure adequate power to address the hypothesis being tested?

-Were correct statistical analysis used to support conclusions?

-Are there concerns about ethical or regulatory requirements being met?

Reviewer #1: Yes. The mixed methods approach in this study is well-designed and described and very appropriate to address their research questions. Although my expertise is in qualitative methods, I believe they have an excellent sample size for both the qualitative and quantitative analysis.

Reviewer #2: - Yes

- Based on the aim of the study stated at line 83-85, the quantitative method is sufficient for primary objective. I suggest the qualitative method for secondary outcomes

- Has the semi structured interview question got trial before? How to prevent bias with those open-ended question method? Line 130-133

- sample size is sufficient 

- statistical analysis is correct for quantitative data, but how you manage the quantitative data for present the results?

- no concern about ethical issue

**Results**

-Does the analysis presented match the analysis plan?

-Are the results clearly and completely presented?

-Are the figures (Tables, Images) of sufficient quality for clarity?

Reviewer #1: Yes, the analysis is well-presented and matches the plan. I had one minor note about Table 1: It should be specified that the first numbers are actual N/numbers. At the bottom it just says, “Data are number (%) unless otherwise specified.”

Reviewer #2: - Is the information of line 156-170 the result of the study? if not and you want to provide the comparative data, you can state in the discussion section 

- Line 189-190: Do you mean 197 out of 303 patients who had complete data previously? Or are those 197 from all patients who completed MDT?

- Line 178-181: if you also want to give specific data about the child, you should state it at the method 

- For table 4: how do you count the percentage? Is the absent + present disability equal to the total of the patient?

- For the interview result, can you present it in the graphics or something to make more attractive?

**Conclusions**

-Are the conclusions supported by the data presented?

-Are the limitations of analysis clearly described?

-Do the authors discuss how these data can be helpful to advance our understanding of the topic under study?

-Is public health relevance addressed?

Reviewer #1: The Discussion and Conclusions are strongly supported by the data, i feel. I think the authors do a good job discussing the limitation that the data were collected at a referral center and so some of the disability data may not be representative of people affected by leprosy throughout Ethiopia, but the I feel that this limitation does not detract from their important findings, and the interviews with HCWs at referral centers was necessary to assess the skills and problems with HCWs at frontline public health facilities.

Reviewer #2: - Line 403-406: you might complete the statement with the decline of disability data at completion of MDT 

- The limitation analysis is appropriate 

- Yes

- Yes

**Editorial and Data Presentation Modifications?**

Reviewer #1: Minor editorial notes: 

Line 39: enragement—maybe “enlargement”? 

Lines 75-79: I think there is a word or two missing here. I think maybe “to” needs to go before “estimate,” or you could say “for early case detection and estimation of the delay” 

Line 138: “on daily basis”—should be “on a daily basis” 

Line 160: “Eastern Mediterranean”—I see that WHO includes 22 countries (including parts of North African and the Horn of Africa) in their Regional Office for the Eastern Mediterranean but not Ethiopia. You might give more information here about how Eastern Mediterranean is defined though. 

Line 162: “increment” –should this be “increase” or maybe “incremental change”? 

Line 178: “child” should be “children” 

Line 215: “reference source not found” –should be resolved here or deleted? 

Line 338: “older” –should be just “age” or “being older”

Reviewer #2: -accept with minor revision-

**Summary and General Comments**

Reviewer #1: I think this article is a very important contribution and contains important suggestions (for WHO and the global leprosy community) for the future of documentation of leprosy disability and for taking steps to prevent and address disability, including the importance of training for HCW who manage cases in public healthcare facilities, the recommendation that GD1 disabilities be reported to WHO, and that collection of data on disability at diagnosis only is flawed, as it might miss the worsening disability that some people experience after starting treatment or after MDT is complete. I think it's also important that you mention the attention to disability in policy related to the 2030 goals, which seem increasingly less realistic. 

I have included minor editorial notes in a box above. My only other suggestion is that you elaborate more on leprosy reaction. You mention that some healthcare workers talked about patients discontinuing medications when they develop leprosy reactions. You could have a brief explanation of what leprosy reactions are in the section on clinical aspects of leprosy, and then I think it is important to highlight how easy it is for patients to misunderstand what is happening when they develop symptoms of leprosy reaction after starting or while taking MDT, especially if HCWs have not really explained this to them in advance. This could also be mentioned briefly in the discussion section as well and of course also related to the importance of training for all HCWs working with people affected by leprosy.

Reviewer #2: (No Response)

PLOS authors have the option to publish the peer review history of their article (what does this mean?). If published, this will include your full peer review and any attached files.

Reviewer #1: Yes: Cassandra White

Reviewer #2: No
---

## [Editor Report · Decision Letter 1]

24 Nov 2024

Dear Dr. Tegene,

We are pleased to inform you that your manuscript 'Relevance of reporting leprosy related disability at the completion of multi drug therapy: A 5-year retrospective analysis of disability in persons affected by leprosy at ALERT Hospital Ethiopia' has been provisionally accepted for publication in PLOS Neglected Tropical Diseases.

Best regards,

Claudio Guedes Salgado, PhD

Academic Editor

Mathieu Picardeau

Section Editor

Shaden Kamhawi

co-Editor-in-Chief

Paul Brindley

co-Editor-in-Chief

---

## [Editor Report · Acceptance letter]

4 Dec 2024

Dear Dr. Tegene,

We are delighted to inform you that your manuscript, "Relevance of reporting leprosy related disability at the completion of multi drug therapy: A 5-year retrospective analysis of disability in persons affected by leprosy at ALERT Hospital Ethiopia," has been formally accepted for publication in PLOS Neglected Tropical Diseases.

Best regards,

Shaden Kamhawi

co-Editor-in-Chief

Paul Brindley

co-Editor-in-Chief
